# Contribution of [^18^F]FET PET in the Management of Gliomas, from Diagnosis to Follow-Up: A Review

**DOI:** 10.3390/ph17091228

**Published:** 2024-09-18

**Authors:** Jade Apolline Robert, Arthur Leclerc, Mathilde Ducloie, Evelyne Emery, Denis Agostini, Jonathan Vigne

**Affiliations:** 1CHU de Caen Normandie, UNICAEN, Department of Nuclear Medicine, Normandie Université, 14000 Caen, France; robert-ja@chu-caen.fr (J.A.R.);; 2Department of Neurosurgery, Caen University Hospital, 14000 Caen, France; 3Caen Normandie University, ISTCT UMR6030, GIP Cyceron, 14000 Caen, France; 4Department of Neurology, Caen University Hospital, 14000 Caen, France; 5Centre François Baclesse, Department of Neurology, 14000 Caen, France; 6CHU de Caen Normandie, UNICAEN Department of Pharmacy, Normandie Université, 14000 Caen, France; 7Centre Cyceron, Institut Blood and Brain @ Caen-Normandie, Normandie Université, UNICAEN, INSERM U1237, PhIND, 14000 Caen, France

**Keywords:** neuro-oncology, glioma, fluoroethyltyrosine (FET), PET, nuclear medicine

## Abstract

Gliomas, the most common type of primary malignant brain tumors in adults, pose significant challenges in diagnosis and management due to their heterogeneity and potential aggressiveness. This review evaluates the utility of O-(2-[^18^F]fluoroethyl)-L-tyrosine ([^18^F]FET) positron emission tomography (PET), a promising imaging modality, to enhance the clinical management of gliomas. We reviewed 82 studies involving 4657 patients, focusing on the application of [^18^F]FET in several key areas: diagnosis, grading, identification of IDH status and presence of oligodendroglial component, guided resection or biopsy, detection of residual tumor, guided radiotherapy, detection of malignant transformation in low-grade glioma, differentiation of recurrence versus treatment-related changes and prognostic factors, and treatment response evaluation. Our findings confirm that [^18^F]FET helps delineate tumor tissue, improves diagnostic accuracy, and aids in therapeutic decision-making by providing crucial insights into tumor metabolism. This review underscores the need for standardized parameters and further multicentric studies to solidify the role of [^18^F]FET PET in routine clinical practice. By offering a comprehensive overview of current research and practical implications, this paper highlights the added value of [^18^F]FET PET in improving management of glioma patients from diagnosis to follow-up.

## 1. Introduction

Gliomas represent the majority of primary malignant brain tumors in adults, with a yearly incidence of approximately 6 per 100,000 in Europe [1]. They are categorized according to the World Health Organization (WHO) classification into grades ranging from 1 to 4 depending on their malignancy [2]. Glioblastoma, the most aggressive and common type of glioma, remains incurable with an almost systematic progression within the year and a median survival of 14.6 months despite optimal treatment [3].

In high-grade tumors, treatment usually consists of maximal resection of the tumor (if feasible) followed by chemotherapy and radiotherapy depending on tumor grade and analysis of molecular markers (i.e., 1p/19q codeletion, IDH mutation, and MGMT promoter methylation) [4]. Treatment of grade 4 gliomas, the same since 2005, is based on the so-called “Stupp protocol”, which includes concomitant radiochemotherapy with Temozolomide [3].

Patients’ monitoring consists of MRI before and after treatment with periodic follow-up. An increase in enhancing areas is considered suspect of recurrence according to the Response Assessment in Neuro-Oncology (RANO) criteria but is not specific [5]. Indeed, frequent post-radiation changes such as pseudoprogression and radionecrosis can cause the same type of suspicious gadolinium-enhancing lesion.

Pseudoprogression typically occurs several weeks up to months (often less than 3 months) after completion of radiotherapy. This phenomenon is responsible for a transitory worsening of MR imaging with an increased contrast enhancement area, resolving without changes in treatment on subsequent MRI scans. There is generally no symptom associated.

Radionecrosis is a severe reaction to radiotherapy, which generally occurs later, months to several years after radiation therapy. MRI findings involve a space-occupying necrotic lesion with a mass effect, which can cause neurological dysfunction.

MRI changes can also be induced by treatments such as corticosteroids, antiangiogenic therapy, or immunotherapy. 

For these reasons, there is a need to find other reliable methods to differentiate glioma recurrence from treatment-related changes, given the different managements of these two processes. 

Different MRI techniques have been implemented in this indication, such as diffusion weighted imaging (DWI) [6], perfusion-weighted imaging (PWI) [7], and magnetic resonance spectroscopy (MRS) [8].

In nuclear medicine, positron emission tomography using 2-deoxy-2-[^18^F]fluoro-D-glucose ([^18^F]FDG) has already proven itself in oncology imaging and has become common practice in numerous pathologies. However, its physiologically high brain metabolism and increased uptake in inflammatory lesions make it difficult to appreciate tumor uptake [9]. 

Radiolabeled amino acids are preferred in neuro-oncology due to low uptake in normal brain tissue contrasting with increased uptake in neoplastic processes, resulting in a better signal-to-noise ratio [10]. 

The most widely used amino acid tracers for PET are [^11^C-methyl]-methionine ([^11^C]MET), O-(2-[^18^F]fluoroethyl)-L-tyrosine ([^18^F]FET), and 3,4dihydroxy-6-[^18^F]fluoro-L-phenylalanine ([^18^F]F-DOPA) (Table 1). Their uptake is believed to be driven by an overexpression of the L-type amino-acid transporter (LAT) by brain tumors (Figure 1).


*Detailed Description of different radiolabeled amino acids*


^11^C-Methionine ([^11^C]MET)

Mechanism: [^11^C]MET is an amino acid analog taken up by tumor cells via the L-type amino acid transporter (LAT). It reflects increased protein synthesis, which is often elevated in gliomas.

Advantages: High sensitivity in detecting both low- and high-grade gliomas; more effective in high-grade gliomas [11]. Provides rapid uptake and good contrast between tumor and normal brain tissue. It is particularly effective to detect tumor recurrence [12] and in monitoring therapy response [13].

Disadvantages: The short half-life of ^11^C (about 20 min) necessitates the use of an on-site cyclotron, limiting its use to specialized centers. [^11^C]MET may also accumulate in inflammatory tissues, leading to potential false positives [14].

[^18^F]F-DOPA 

Mechanism: [^18^F]F-DOPA is a precursor to dopamine and is taken up by dopaminergic neurons, with uptake also observed in gliomas due to increased amino acid transport and altered tumor metabolism. It is decarboxylated to dopamine and subsequently trapped in cells.

Advantages: The longer half-life of ^1^⁸F (about 110 min) allows for broader clinical application as it can be transported from off-site production facilities. It has high sensitivity for gliomas [15] and is particularly useful in differentiating between tumor recurrence and radiation necrosis [16].

Disadvantages: Uptake of [^18^F]F-DOPA in inflamed tissues can lead to false-positive results [17]. 

^1^⁸F-Fluoroethyl-L-tyrosine ([^18^F]FET)

Mechanism: [^18^F]FET is an artificial amino acid taken up by glioma cells via LAT, reflecting the increased amino acid transport associated with tumor proliferation.

Advantages: [^18^F]FET has a longer half-life, like ^1^⁸F-DOPA, allowing it to be produced off-site. It has high sensitivity for gliomas, especially high-grade gliomas [18], with low uptake in inflammatory lesions, making it particularly effective in distinguishing tumor recurrence from treatment-induced changes. Additionally, dynamic acquisition allows information on tracer kinetics, particularly useful for tumor grading [19].

Disadvantages: Though it offers high specificity. There is also potential, though reduced, for uptake in inflammatory tissues [20].

While recent meta-analyses report high sensitivity and specificity of both ^1^⁸F-DOPA and [^18^F]FET to differentiate true progression to treatment-related changes, there are still discrepancies in determining the best radiolabeled amino acid [21,22,23].

[^18^F]FET market authorizations have been delivered in Europe recently, enabling its widespread use in hospitals.

Its high efficiency production and its half-life of 110 min allow its transportation to other sites. For these reasons, it is being increasingly used in glioma management in Europe. 

In the present review, we aimed to summarize its performance in different indications in low- and high-grade gliomas.

## 2. Materials and Methods

### 2.1. Search Strategy

The primary literature was searched up to 31 December 2023, using the PubMed database. 

A combination of the search terms «PET», «FET» OR «amino acid» OR «fluoroethyltyrosine» OR «fluoroethylltyrosine», «Glioma» OR «brain tumor», «pediatric», and «neuro-oncology» were used. The screening of abstracts and full-text articles was performed by one reviewer (J.A.R.). 

Inclusion criteria were studies in English, using FET, and in humans with a full text available.

Exclusion criteria included studies that included less than 20 patients, did not report on diagnostic test parameters or metrics representing impact on clinical management decisions and/or survival outcomes, did not give information about histology or tumor grades, and studies that included other malignancies. We also excluded studies that did not include histological confirmation or follow-up.

### 2.2. Data Synthesizing

For each study, the indication, principal author, publication year, study design, number of patients, grade, age, sex, type of imaging modality, test parameter, cut-off used, and their performances were recorded.

## 3. Results

### 3.1. Literature Search

We selected 152 studies according to their title and abstract, but upon full-text review, 70 studies were excluded (Figure 2).

The remaining 82 studies [19,24,25,26,27,28,29,30,31,32,33,34,35,36,37,38,39,40,41,42,43,44,45,46,47,48,49,50,51,52,53,54,55,56,57,58,59,60,61,62,63,64,65,66,67,68,69,70,71,72,73,74,75,76,77,78,79,80,81,82,83,84,85,86,87,88,89,90,91,92,93,94,95,96,97,98,99,100,101,102,103,104] were included in this review, with a total of 4657 patients. Details of these study characteristics can be found in Table 2.

Regarding PET parameters, we noticed a high variability in the determination of tumor region of interest (ROI) with an impact on the subsequent calculation of tumor-to-brain ratios (TBRs). We consequently sorted different TBRs according to the methodology used to obtain them (Table 3) in order to be able to compare their performances and then grouped every PET parameter in Table 4. We signified the change of parameters in the legend of Table 4 by writing the name of the parameter used in the table and the name of the original parameter(s) corresponding to this approach.

### 3.2. Diagnosis

Four prospective studies [24,25,26,27] evaluated the performance of [^18^F]FET PET in patients with cerebral lesions suspicious of glioma. Each study chose a different method of TBR determination to detect glioma tissue with a threshold of 1.6 in two of them [26,27], resulting in a sensitivity of 88 to 92% and a specificity of 81 to 88%. 

### 3.3. Grading

Thirteen studies [19,28,29,30,31,32,33,34,35,36,37,38,39] evaluated the performance of [^18^F]FET PET in glioma grading. Most studies aimed at differentiating low-grade gliomas (LGGs) from high-grade gliomas (HGGs). Multiple TBR methods were used, with a predominance of maximum tumor-to-brain ratio (TBR_max_) with sensitivity and specificity ranging from 67 to 92% and 61 to 85%, respectively. Dynamic parameters and notably tumor-activity curves (TAC) had better performance, with a sensitivity of 73 to 96% and a specificity of 63 to 100%.

Notably, one study by Lohmann et al. [31] chose to supplement dynamic imaging from 0 to 50 min post-injection (p.i.) with an additional acquisition from 70 to 90 min p.i. The goal was to compare conventional dynamic imaging to dual-time-point imaging: one acquisition from 20 to 40 min p.i. and a delayed second acquisition from 70 to 90 min p.i. Mean tumor-to-brain ratio (TBR_mean_) change and TAC achieved similar accuracy of 81% and 83%, respectively.

### 3.4. IDH Status Determination

Six retrospective studies [34,40,41,42,43,44] evaluated the performance of [^18^F]FET PET in IDH status determination. Static parameters’ significancy was variable depending on the studies, whereas dynamic ones (Slope, Time-to-peak (TTP), TAC) always showed significant differences between IDH mutated and IDH wild-type groups with an accuracy of around 73%.

### 3.5. Prediction of Oligodendroglial Components

Two studies [38,44] reported on the performance of [^18^F]FET PET to determine the presence of oligodendroglial tumor components. Every static parameter tested was significant. Tumor-to-brain ratios showed good sensitivity, but specificity did not exceed 65%.

There were no dynamic parameters studied.

### 3.6. Guided Resection or Biopsy

Four studies [45,46,47,48] tested the addition of [^18^F]FET PET to better detect tumor tissue for resection or biopsy. In a study by Ewelt et al. [47], results were separated according to glioma grades (LGG vs. HGG), showing better tissue detection in high-grade glioma with sensitivity and specificity of 88% and 46%. Sensitivity was higher than those of MRI and 5-ALA-fluorescence, with a specificity being the lowest. Combining different modalities did not improve results compared to those of 5-ALA-fluorescence alone (sensitivity of 71% and specificity of 92%).

### 3.7. Detection of Residual Tumor

Two studies [49,50] aimed at detecting residual tumor tissue after surgery.

Buchmann et al. [49] also aimed to assess whether performing [^18^F]FET PET after 72 h after neurosurgery had an influence, as it is the case with MRI. Indeed, postoperative MRI after 72 h can lead to falsification of results because of inflammatory reactions. This study found higher sensitivity of PET using a TBR > 1.6 compared to MRI and no influence of timing of [^18^F]FET PET imaging. 

### 3.8. Guided Radiotherapy

Studies [51,52,53,54,55,56] used the TBR threshold of 1.6 to define the tumor volume to be irradiated. This PET-based volume was increased compared to the MRI-based volume commonly used.

One study (Harat et al. [54]) reported 74% of failures inside primary gross tumor volume (GTV) PET volumes, with no solitary progressions inside the MRI-defined margin +20 mm but outside the GTV PET detected.

### 3.9. Detection of Malignant Transformation in Low-Grade Gliomas

Three studies [57,58,59] evaluated the use of [^18^F]FET PET to detect differences between non-transformed LGGs and LGGs that had transformed to high-grade gliomas. Two studies found a good detection value of both static and dynamic parameters in this indication, whether by comparing to baseline or by using parameter thresholds.

The remaining study (Bashir et al. [59]) did not find significant differences when considering all patients. After excluding the oligodendroglial subgroup, however, a significant difference was observed between non-transformed and transformed LGGs when combining [^18^F]FET parameters. The best result was observed with a combined analysis of TBR_max_ > 1.6 and TAC with a plateau or decreasing pattern (sensitivity of 75% and specificity of 83%).

### 3.10. Recurrence vs. Treatment-Related Changes 

Twenty studies [60,61,62,63,64,65,66,67,68,69,70,71,72,73,74,75,76,77,78,79] evaluated the performance of [^18^F]FET PET in the differentiation of recurrence from treatment-related changes.

The majority of studies included patients treated with multiple modalities (such as operation, chemotherapy, and radiotherapy) who had a suspected tumor recurrence or progression as revealed by follow-up MRI. High-grade gliomas represented 87% (992/1141) of tumors.

Most studies used static parameters TBR_max_ and TBR_mean_ along with dynamic parameters TTP and Slope.

TBR_max_ was significant in 13 studies with thresholds between 1.64 and 3.69. TBR_mean_ significantly differentiated recurrence from pseudoprogression in 11 studies. The thresholds used varied from 1.8 to 2.31. Accuracy of TBR_max_ and TBR_mean_ was comparable.

Dynamic parameters, when combined with static ones, allowed to increase diagnostic accuracy in some studies such as Werner et al. [68] and Galldiks et al. [78]. In Werner et al., TBRs alone had a diagnostic accuracy of 83%, which increased to 90% and 93% when combined with TTP and Slope, respectively. This finding was not supported by other studies, such as Werner et al. [66] and Galldiks et al. [67].

### 3.11. Prognosis and Treatment Response Evaluation

Twenty-eight studies [39,43,61,80,81,82,83,84,85,86,87,88,89,90,91,92,93,94,95,96,97,98,99,100,101,102,103,104] evaluated the performance of [^18^F]FET PET in prognosis and treatment response evaluation.

Prognostic parameters can be extracted before, during, and after treatment. For example, Pyka et al. [93] studied patients with untreated, first-diagnosed gliomas and were able to predict tumor recurrence, with dynamic parameters showing better results than static ones, especially in the low-grade subgroup.

Overall, static parameters tended to not reach significance, whereas dynamic ones such as TTP and TAC demonstrated better results. TTP was the best parameter in two studies (Pyka et al. [93] and Bauer et al. [95]) with AUCs of 0.848 and 0.90, respectively.

Many studies also decided to use biological tumor volume (BTV), often determined by an autocontouring process using a TBR threshold of 1.6. Every study used a different cut-off when considering absolute values, and half of them did not reach significance. Three studies [82,87,94] opted for a BTV change after the initiation of chemotherapy to separate responders (relative change ≤ 0%) from non-responders (relative change > 0%). Two of them examined patients at first diagnosis and the third one at recurrence. These studies found a decreasing BTV to predict a significantly longer progression-free survival and to be associated with prolonged overall survival.

### 3.12. Radiomics

Radiomic parameters were used by 1 study, for grading [39] (grade 3 vs. 4), 2 studies in IDH status determination [40,41], 2 studies in the differentiation of recurrence vs. pseudoprogression [69,76], and 2 studies for prognosis [39,89].

Different textural features showed good performance in each study, and the combination of standard PET parameters with textural features could improve results, for example in IDH genotype determination, as shown by Lohmann et al. [41]. Combination of the dynamic parameter Slope with the radiomic feature SZHGE slightly increased diagnostic accuracy to 81% vs. 80% with Slope alone.

## 4. Discussion/Conclusions

This review proposes an up-to-date summary of PET performance in glioma management using O-(2-[^18^F]fluoroethyl)-L-tyrosine. The homogenization of PET tumor-to-brain ratios according to the determination of the different regions of interest allowed to truly compare their sensibility, specificity, AUC, and accuracy.

[^18^F]FET can be useful in every step of glioma management, from diagnosis to suspicion of recurrence.

The ability to discriminate tumor tissue from healthy brain tissue is helpful in diagnosis, to guide a surgical procedure or radiotherapy, and to detect the presence of a residue after surgery. Most studies agree on a TBR threshold > 1.6 to delineate tumor extent. 

Different thresholds of tumor-to-brain ratio are also useful to predict histological characteristics (low vs. high grade, malignant transformation of a low-grade glioma, and oligodendroglial components), to differentiate post-treatment changes from a true recurrence, and to extract prognostic parameters and assess treatment response.

It is important to note that while many studies used static parameters TBR_max_ and TBR_mean_, the definition of these ratios differs depending on the article. For example, the ratio between the mean standard uptake value (SUV_mean_) of a 16 mm ROI centered on the maximal tumor uptake and the SUV_mean_ of a contralateral background ROI, named TBR_16mm_ in this review, can be called TBR_mean_ in a study (Verger et al. [64]) and TBR_max_ in another (Galldiks et al. [78]). 

Kertels et al. [63] expressed the need to use comparable approaches to be able to obtain relevant and reliable results. Despite the absence of a significant difference between methods chosen, approaches focusing on voxels with the highest uptake tended to perform superior.

Dynamic acquisition also adds valuable information with parameters such as TTP, TAC, or Slope and should be preferred. An interesting alternative proposed by Lohmann et al. [31] is dual-time point imaging, allowing to reduce costs due to higher patient throughput and imaging time.

Relatively new tools are also available, such as radiomics and hybrid PET/MR imaging, and could be of great interest in the future. The use of hybrid PET/MR is set to increase in neuro-oncology and could improve performance, as suggested by Lohmann et al. [41] concerning radiomics.

Joint EANM/EANO/RANO practice guidelines [9] published in 2018 summarized methods and cut-off values in different clinical situations concerning radiolabeled amino acids and [^18^F]FDG. It is of importance to note that the studies used to extract these guidelines are often retrospective and/or based on small effectives.

At the beginning of the year, Albert et al. [105] published the first version of PET RANO criteria in an effort to facilitate the structured implementation of PET imaging into clinical research and, ultimately, clinical routine.

The principal limitation of this review is the methodology used and the fact that many of the included studies are also retrospective and do not reflect clinical practice. Additionally, none of the studies included focused on pediatric gliomas, probably because of the limited number of patients in the available research.

While [^18^F]FET is becoming an important tracer in neuro-oncology, [^18^F]F-DOPA also showed good results and should not be overlooked. A recent meta-analysis and systematic review compared [^18^F]F-DOPA and [^18^F]FET for differentiating treatment-related change from true progression (Yu et al. [21]) and found that [^18^F]F-DOPA seems to demonstrate superior sensitivity and similar specificity to [^18^F]FET. Nevertheless, [^18^F]F-DOPA PET results were obtained from studies with limited sample sizes.

There is a need to pursue research with prospective, multicentric studies to be able to standardize imaging analysis and define the use of technological advancements such as hybrid PET/MRI imaging and radiomics and to compare [^18^F]FET with existing radiopharmaceuticals such as [^18^F]F-DOPA head-to-head comparisons.

## Figures and Tables

**Figure 1 pharmaceuticals-17-01228-f001:**
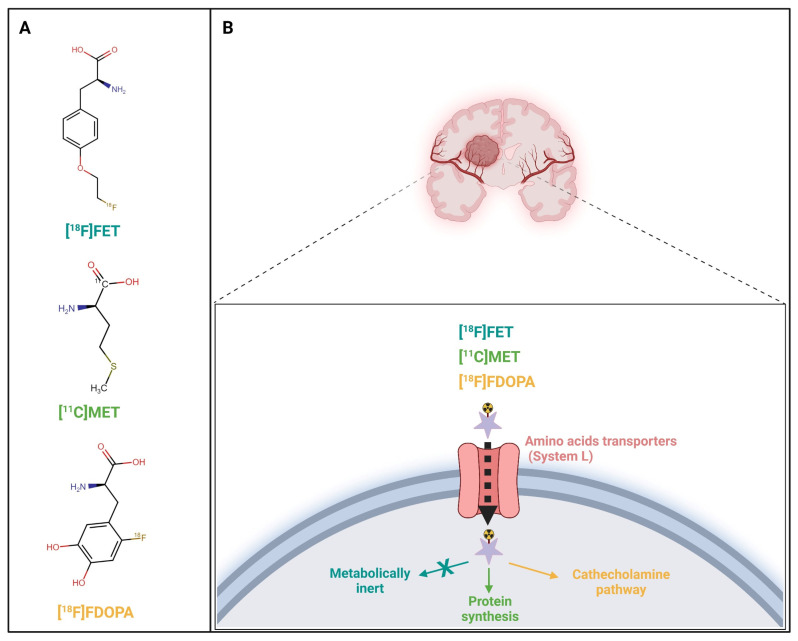
Radiolabeled amino acids O-(2-[^18^F]fluoroethyl)-L-tyrosine ([^18^F]FET), [^11^C-methyl]-methionine ([^11^C]MET), and L-3,4-dihydroxy-6-[^18^F]fluoro-phenyl-alanine ([^18^F]FDOPA) metabolic pathways. Molecular structures (**A**) and associated uptake mechanism (**B**) of each radiolabeled amino acid. Created with BioRender.com.

**Figure 2 pharmaceuticals-17-01228-f002:**
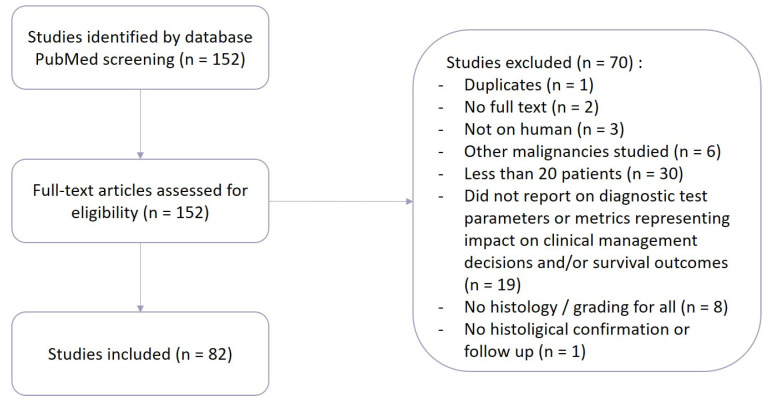
Flowchart of the literature selection.

**Table 1 pharmaceuticals-17-01228-t001:** Comparative table of different radiolabeled amino acids.

Aspect	[^11^C]MET	[^18^F]F-DOPA	[^18^F]FET
Radiotracer Type	Amino acid analog	Amino acid precursor	Amino acid analog
Mechanism of Uptake	Uptake via L-type amino acid transporter (LAT) into tumor cells with high protein synthesis.	Uptake via amino acid transport (LAT) is overexpressed in tumor cells. Converted into dopamine in dopaminergic neurons.	Uptake via LAT, reflecting increased amino acid transport correlated to tumor proliferation.
Half-Life	20 min	110 min	110 min
Production	Requires on-site cyclotron due to short half-life.	Can be produced off-site, longer shelf life.	Can be produced off-site, longer shelf life.
Sensitivity in Gliomas	High sensitivity, more effective in detecting high-grade gliomas.	High sensitivity in detecting glioma.	High sensitivity, more effective in detecting high-grade gliomas.
Specificity in Gliomas	Moderate specificity, possible uptake in inflammatory lesions.	High specificity, with potential uptake in inflammatory tissues.	High specificity, with less non-specific uptake in inflammatory tissues compared to [^11^C]MET.
Advantages	Rapid uptake, good lesion contrast.	Longer half-life allows broader clinical application.	Longer half-life allows broader clinical application. Dynamic acquisition allows additional information on tracer kinetics, particularly useful for tumor grading.
Disadvantages	Short half-life limits use to facilities with a cyclotron, potential uptake in inflammation.	May have false positives in inflamed tissues. High physiologic uptake in the basal ganglia.	Potential uptake in inflammatory lesions but less than [^11^C]MET.
Clinical Application	Primarily used in facilities with a cyclotron, used to detect tumor recurrence and in monitoring the response to therapy.	Mostly used for differentiating tumor recurrence from necrosis, especially in high-grade gliomas.	Widely used for differentiating high-grade glioma early and late progression from radiation effects.

**Table 2 pharmaceuticals-17-01228-t002:** Characteristics of the 82 included studies. §: did not reach significance, &: did not reach significance after Bonferroni multiple-test correction, #: significance not available.

Indication	Author, Year	Reference	Design	Number of Patients	Grade	Mean Age	Sex	Imaging Modality	Parameters	Optimal Cut-Off	Sensitivity	Specificity	AUC	Accuracy
Diagnosis														
	Pauleit et al., 2009	[24]	Prospective	52	Not glioma:9	46	36 M 16 F	PET	Lmean/B #	-				
					Grade 2:22				Lmax/B #	-				
					Grade 3:12				Visual grading system #	-				
					Grade 4:9									
	Mauler et al., 2023	[25]	Prospective	30	Not glioma:6	48	16 M 14 F	PET	^18^F-FETn uptake	1.4 x background	76%	80%	0.89	78%
					Grade 2:7			MRI	Cho/NAAn	2.16	59%	83%	0.81	71%
					Grade 3:7									
					Grade 4:10									
	Floeth et al., 2005	[26]	Prospective	50	Not glioma:16	44	21 M 29 F	PET	FET lesion/brain ratio	1.6	88%	88%		-
					Grade 1:2			MRI	Gd enhancement	-	44%	69%		68%
					Grade 2:13				NAA/Cho ratio	0.7	100%	81%		-
					Grade 3:14									
					Grade 4:5									
	Pauleit et al., 2005	[27]	Prospective	28	Not glioma:5	42	9 M 19 F	PET	FET ratio	1.6	92%	81%		-
					Grade 1:2			MRI	T1 ratio	1.0	85%	12%		-
					Grade 2:7				Gd-T1 ratio	1.0	38%	96%		-
					Grade 3:12				FLAIR ratio	1.0	96%	4%		-
					Grade 4:2				T1/Gd-T1/FLAIR ratio	-	96%	53%		68%
								PET/CT + MRI	FET/T1/Gd-T1/FLAIR ratio	-	93%	94%		94%
Grading (LGG vs. HGG)														
	Jeong and Lim, 2012	[28]	Prospective	20	Grade 2:3	52	13 M 7 F	PET	SUVmax	-				
					Grade 3:2				TNR	-				
					Grade 4:15									
	Verger et al., 2017	[29]	Retrospective	72	Grade 1:1	49	42 M 30 F	PET	TBRmax	2.62	82%	68%	0.83	78%
					Grade 2:21				TBRmean	1.69	82%	68%	0.80	78%
					Grade 3:25				TTP	30 min	54%	91%	0.78	65%
					Grade 4:25				Slope	−0.03 SUV/h	64%	91%	0.78	72%
								PWI rCBF	TBRmax	1.51	64%	64%	0.74	64%
									TBRmean	0.69	62%	59%	0.66	61%
								PWI rCBV	TBRmax	1.80	88%	72%	0.81	83%
									TBRmean	1.14	72%	77%	0.80	74%
								PWI MTT	TBRmax §	1.16	64%	50%	0.58	60%
									TBRmean §	0.98	54%	36%	0.43	49%
	Lopez et al., 2015	[30]	Prospective	23	No-grade:2	56	18 M 5 F	PET	UR	3.0				
					Grade 1:1									
					Grade 2:7									
					Grade 3:2									
					Grade 4:11									
	Lohmann et al., 2015	[31]	Prospective	36	Grade 2:12	49	19 M 17 F	PET	TBRmean §	2	83%	58%	0.65	75%
					Grade 3:8				∆TBRmean 20–40 min/70–90 min	−8%	83%	75%	0.85	81%
					Grade 4:16				TTP	35 min	58%	92%	0.76	69%
									Kinetic pattern	II/III	88%	75%	-	83%
	Calcagni et al., 2011	[32]	Prospective	32	Grade 1:3	41	21 M 11 F	PET	TAC #	I/II vs. III	73%	100%		87%
					Grade 2:14				Early SUV	2.32	73%	71%		72%
					Grade 3:11				Middle SUV §	-	-	-		-
					Grade 4:4				Late SUV §	-	-	-		-
									e-m ratio	0.93	93%	94%		94%
									e-l ratio	0.95	87%	88%		87%
									Tpeak	25 min	87%	100%		94%
									SoD	0.5	93%	82%		87%
									Logistic regression using Early SUV + SoD §	50%	93%	100%		97%
	Albert et al., 2016	[33]	Retrospective	314	Grade 1:3	49	181 M 133 F	PET	TBRmax (20–40 min)	2.7	67%	78%		70%
					Grade 2:128				TBRmax (0–10 min)	2.8	76%	79%		76%
					Grade 3:95				TBRmax (5–15 min)	2.7	78%	76%		77%
					Grade 4:88				TBRmax (5–20 min)	2.6	80%	74%		76%
									TBRmax (10–30 min)	2.5	75%	75%		74%
									Kinetic pattern #	Decreasing	90%	66%		80%
	Pöpperl et al., 2007	[19]	Prospective	54	Grade 2:15	49	30 M 24 F	PET	SUVmax/BG	2.58	71%	85%	0.798	
					Grade 3:21				SUV90 10–60 min	0.20	94%	100%	0.969	
					Grade 4:18				SUV90 15–60 min	−0.41	94%	100%	0.965	
Grade 2/3 vs. grade 4	Hua et al., 2021	[34]	Retrospective	58	Grade 2:33	42	37 M 21 F	PET	TBRmax	2.67	92%	61%	0.824	67%
					Grade 3:13				TBRpeak	2.35	92%	61%	0.832	67%
					Grade 4:12				TBRmean	2.31	58%	93%	0.791	86%
									COV	27.21	58%	91%	0.808	84%
									HI	1.77	67%	87%	0.826	83%
									MTV	20.13	75%	80%	0.801	79%
									TLU	50.93	75%	83%	0.841	81%
									SUVsd	0.45	67%	87%	0.816	83%
									TBRmax + SUVsd + TBRmean	-	75%	85%	0.850	83%
									HI + SUVsd + MTV	-	75%	83%	0.848	81%
									HI + SUVsd + TLU	-	75%	84%	0.848	81%
	Kunz et al., 2011	[35]	Prospective	55	Grade 2:31	44	33 M 22 F	PET	TAC	Increasing vs. decreasing	96%	94%		
					Grade 3:22			MRI	Tumor volume §	-	-	-		
					Grade 4:2									
Grade 2/3 vs. grade 4	Röhrich et al., 2018	[36]	Retrospective	44	Grade 2:10	53	-	PET	TAC #	LGG-like vs. mixed vs. HGG-like	-	-	-	
					Grade 3:13				SUVmax/BG	-	-	-	-	
					Grade 4:21				TTP §	-	-	-	-	
									Relative K1	-	85%	60%	0.766	
									Relative K2 §	-	-	-	-	
									Relative K3 §	-	-	-	-	
									Relative FD	-	67%	78%	0.716	
									SUVmax/BG + TTP	-	-	-	0.745	
									SUVmax/BG + TTP + relative K1 + relative FD	-	-	-	0.799	
	Jansen et al., 2012	[37]	Retrospective	127	No tumor:7	46	72 M 55 F	PET	TAC #	Increasing vs. decreasing	95%	72%		
					Grade 1:4				FET uptake #	Reduced vs. normal vs. increased	-	-		
					Grade 2:69				FET uptake pattern §	Inhomogeneous vs. diffuse vs. focal	-	-		
					Grade 3:42				SUVmax/BG §	-	-	-		
					Grade 4:5				SUVmean/BG §	-	-	-		
									BTV §	-	-	-		
grade 2 vs. 3	Jansen et al., 2012	[38]	Prospective	144	Grade 2:79	45	84 M 60 F	PET	TAC #	Decreasing	88%	63%		
					Grade 3:65				SUVmax/BG §	-	-	-		
									BTV §	-	-	-		
									SUVtotal/BG §	-	-	-		
									SUVmean/BG §	-	-	-		
grade 3 vs. 4	Pyka et al., 2016	[39]	Retrospective	113	Grade 3:26	59	43 M 70 F	PET	TBRmax §	2.74			0.614	
					Grade 4:87				TBRmean	1.68			0.644	
									MTV	19.7			0.710	
									TLU	46.2			0.704	
									Textural parameters:					
									Coarseness	0.607			0.757	
									Contrast	0.203			0.775	
									Busyness	1.12			0.737	
									Complexity	0.069			0.633	
									Combined	2.05			0.830	
IDH status determination														
	Hua et al., 2021	[34]	Retrospective	58	Grade 2:33	42	37 M 21 F	PET	TBRmax	2.21	48%	87%	0.658	72%
					Grade 3:13				TBRpeak §	2.15	57%	73%	0.638	67%
					Grade 4:12				TBRmean §	1.84	62%	68%	0.633	66%
									COV	8.85	52%	76%	0.650	67%
									HI	1.26	48%	87%	0.676	72%
									MTV	19.48	90%	46%	0.660	62%
									TLU	28.95	81%	57%	0.698	66%
									SUVsd	0.11	47%	57%	0.710	66%
									TBRmax + SUVsd + TBRmean	-	76%	84%	0.821	81%
									HI + SUVsd + MTV	-	86%	81%	0.804	83%
									HI + SUVsd + TLU	-	76%	84%	0.799	81%
	Zhou et al., 2021	[40]	Retrospective	58	Grade 2:31	-	26 M 22 F	PET	SUVSD	0.23	-	-	-	-
					Grade 3:14				TLU §	-	-	-	-	-
					Grade 4:13				MTV §	-	-	-	-	-
									TBRmax §	-	-	-	-	-
									TBRmean §	-	-	-	-	-
									TBRpeak §	-	-	-	-	-
									Midline involvement	Yes vs. no	-	-	-	-
									Simple predictive model	-	85%	71%	0.786	76%
									Radiomics models:					
									PET-Rad model	-	80%	74%	0.812	76%
								CT	CT-Rad model	-	85%	76%	0.883	79%
								PET/CT	PET/CT-Rad model	-	85%	87%	0.912	86%
	Lohmann et al., 2018	[41]	Retrospective	84	Grade 2:7	54	50 M 34 F	PET	TBRmean	1.68	12%	100%	0.66	73%
					Grade 3:26				TBRmax §	2.07	8%	100%	0.59	71%
					Grade 4:51				TTP	45 min	27%	93%	0.75	73%
									Slope	0.30 SUV/h	58%	90%	0.79	80%
									Slope + Radiomic feature SZHGE	-	54%	93%	-	81%
									Radiomic features:					
									SkewnessH §	-	31%	90%	0.53	71%
									LRHGE §	-	8%	100%	0.52	71%
	Verger et al., 2018	[42]	Retrospective	90	Grade 2:16	51	55 M 35 F	PET	TBRmean	1.85	44%	92%	0.73	69%
					Grade 3:27				TBRmax	2.15	56%	77%	0.68	67%
					Grade 4:47				TTP	25 min	86%	60%	0.75	72%
									Slope	−0.26 SUV/h	81%	60%	0.75	70%
									TBRmean + TBRmax	1.85 and 2.15	44%	91%	-	69%
									TTP + Slope	25 min and −0.26 SUV/h	77%	70%	-	73%
									TBRmean + TTP	1.85 and 25 min	40%	96%	-	69%
									TBRmax + TTP	2.15 and 25 min	51%	94%	-	73%
									TBRmean + Slope	1.85 and −0.26 SUV/h	40%	94%	-	68%
									TBRmax + Slope	2.15 and −0.26 SUV/h	47%	91%	-	70%
	Blanc-Durand et al., 2018	[43]	Retrospective	37	Grade 1:3	45	23 M 14 F	PET	TBRmax		-	-		
					Grade 2:15				TBRmean		-	-		
					Grade 3:14				TTP		-	-		
					Grade 4:5				Slope		-	-		
									TAC	Centroid #1 vs. centroid #3	-	-		
	Bette et al., 2016	[44]	Retrospective	65	Grade 1:11	38	36 M 29 F	PET	TBR #	1.3	89%	36%		
					Grade 2:54				TBR #	1.6	71%	53%		
									TBR #	2.0	57%	68%		
									TBRmax §	-	-	-		
Prediction of oligodendroglial components														
	Jansen et al., 2012	[38]	Prospective	144	Grade 2:79	45	84 M 60 F	PET	SUVmax/BG	2.6	70%	72%		
					Grade 3:65				BTV	4.0	71%	69%		
									SUVmean/BG	2.1	61%	59%		
									SUVtotal/BG	6.9	75%	66%		
	Bette et al., 2016	[44]	Retrospective	65	Grade 1:11	38	36 M 29 F	PET	TBR #	1.3	100%	23%		
					Grade 2:54				TBR #	1.6	93%	48%		
									TBR #	2.0	86%	65%		
									TBRmax	-	-	-		
Guided resection/biopsy														
	Ort et al., 2021	[45]	Retrospective	30	Grade 3:5	59	19 M 11 F	PET	BTV	1 cm^3^				
					Grade 4:25									
	Floeth et al., 2011	[46]	Prospective	30 patients/38 biopsies	Grade 2:17	43	20 M 10 F	PET	TBR	1.6				
					Grade 3:19			MRI	Gd-DTPA enhancement	-				
					Grade 4:2			5-ALA-fluorescence	Fluorescent areas	-				
	Ewelt et al., 2011	[47]	Prospective	30	Grade 2:13	42	20 M 10 F		LGG subgroup:					
					Grade 3:15			PET	Tumor/brain tissue ratio	1.6	54%	12%		
					Grade 4:2			MRI	Gd enhancement	-	8%	36%		
								5-ALA-fluorescence	Fluorescent areas	-	8%	29%		
								PET/MRI	-	-	8%	35%		
								MRI/5-ALA	-	-	8%	41%		
								PET/5-ALA	-	-	8%	29%		
								PET/MRI/5-ALA	-	-	8%	41%		
									HGG subgroup:					
								PET	Tumor/brain tissue ratio	1.6	88%	46%		
								MRI	Gd enhancement	-	65%	92%		
								5-ALA-fluorescence	Fluorescent areas	-	71%	92%		
								PET/MRI	-	-	65%	92%		
								MRI/5-ALA	-	-	59%	92%		
								PET/5-ALA	-	-	71%	92%		
								PET/MRI/5-ALA	-	-	59%	92%		
	Verburg et al., 2020	[48]	Prospective	20	Grade 2:8	-	12 M 8 F	PET	TBR	-	-	-	0.76	
					Grade 4:12			T1G-MRI	-	-	-	-	0.56	
								PET/MRI	ADC + TBR	-	-	-	0.89	
Detection of residual tumor														
	Buchmann et al., 2016	[49]	Retrospective	62	Grade 4:62	61	37 M 25 F	PET	TBR	1.6				
								MRI	Contrast-enhanced tissue areas	-				
	Kläsner et al., 2015	[50]	Prospective	25	Grade 2:4	62	16 M 9 F	PET	Visual uptake	>Background				
					Grade 3:3			MRI	Contrast-enhancement volume	0.175 cm^2^				
					Grade 4:18									
Guided radiotherapy														
	Allard et al., 2022	[51]	Prospective	23	Grade 3:3	59	14 M 9 F	PET	TBRmax #	1.6				
					Grade 4:20				SUVmax #	30%				
									SUVmax #	40%				
									SUVmax #	50%				
									SUVmax #	60%				
									SUVmax #	70%				
									SUVmax #	80%				
									SUVmax #	90%				
								CE-MRI	Visual analysis #	-				
	Munck af Rosenschold et al., 2015	[52]	Prospective	54	Grade 3:19	55	-	PET	TBR #	1.6				
					Grade 4:35			CE-MRI	Visual analysis #	-				
	Fleischmann et al., 2020	[53]	Retrospective	36	Grade 4:36	66	20 M 16 F	PET	TBRmax #	1.6				
								MRI	Visual analysis #					
	Harat et al., 2016	[54]	Prospective	34	Grade 4:34	-	-	PET	FET uptake #	1.6 x SUVmean				
								MRI	Visual analysis #	-				
	Dissaux et al., 2020	[55]	Prospective	30	Grade 3:5	63	20 M 10 F	PET	TBR#	1.6				
					Grade 4:25			MRI	Visual analysis #	-				
	Hayes et al., 2018	[56]	Retrospective	26	Grade 3:5	61	17 M 9 F	PET	TBR #	1.6				
					Grade 4:21			CE-MRI	Visual analysis #	-				
								FLAIR-MRI	Visual analysis #	-				
Detection of malignant transformation in LGG														
	Galldiks et al., 2013	[57]	Prospective	27	Grade 2:27	44	18 M 9 F	PET	TBRmax	∆33%	72%	89%	0.87	78%
									TBRmean	∆13%	72%	78%	0.80	74%
									TTP	∆-6 min	72%	89%	0.78	78%
									Kinetic pattern change	I to II/III	72%	89%	-	78%
									TBRmax + TTP + Kinetic pattern change	∆ + 33% or ∆-6 min or I to II/III	83%	78%	-	81%
								MRI	Contrast enhancement change	-	44%	100%	-	63%
	Unterrainer et al., 2016	[58]	Retrospective	31	Grade 2:26	38	18 M 13 F	PET	TBRmax	2.46	82%	89%	0.92	85%
					Grade 3:5				TTPmin	17.5 min	73%	67%	-	70%
	Bashir et al., 2018	[59]	Retrospective	42 patients/47 PET	Inconclusive:2	41	18 M 24 F	PET	TBRmax §	-	57%	41%	0.476	
					Grade 1:1				TAC §	-	71%	41%	0.549	
					Grade 1/2:1				TTP §	25 min	57%	47%	0.511	
					Grade 2:43				TBRmax + TAC + TTP §	1.6 + II/III + 25 min	65%	58%	0.634	
									TBRmax + TAC§	1.6 + II/III	65%	58%	0.639	
									TBRmax + TTP §	1.6 + 25 min	96%	25%	0.591	
								MRI	Contrast enhancement § (CE)	new area	43%	77%	0.597	
								PET/MRI	TBRmax + TAC + TTP + CE §	-	70%	50%	0.643	
									TBRmax + TAC + CE §	-	52%	75%	0.656	
									TBRmax + TTP + CE §	-	57%	58%	0.620	
Recurrence vs. treatment-related changes														
	Jeong et al., 2010	[60]	Retrospective	32	Grade 2:10	47	12 M 20 F	PET	SUVmax	1.66	87%	100%	0.978	
					Grade 3:8				LNR	2.18	86%	88%	0.940	
					Grade 4:14				LGG subgroup:					
									SUVmax	1.48	88%	89%	0.951	
									LNR	1.64	100%	75%	0.893	
									HGG subgroup:					
									SUVmax	1.66	93%	100%	0.993	
									LNR	2.46	86%	100%	0.964	
	Jansen et al., 2013	[61]	Prospective	33	Grade 3:20	-	-	PET	BTV after 6 months	-				
					Grade 4:13				SUVmax/BG after 6 months	-				
	Puranik et al., 2021	[62]	Retrospective	72	Grade 3:13	-	47 M 25 F	PET	T/Wm	2.65	80%	88%		
					Grade 4:59									
	Kertels et al., 2019	[63]	Retrospective	36	Grade 4:36	54	22 M 14 F	PET	TBRmax	3.69	79%	88%	0.86	
									TBRmax	3.58	64%	100%	0.84	
									TBRmax	3.44	86%	88%	0.86	
									TBRmean	2.31	61%	100%	0.83	
									TBRmean	2.19	71%	88%	0.80	
									TBR16 mm	2.44	82%	75%	0.82	
									TBR10 mm	2.86	86%	75%	0.81	
									TBR90%	3.23	71%	100%	0.85	
									TBR80%	3.08	82%	88%	0.88	
									TBR70%	2.72	86%	88%	0.87	
	Verger et al., 2018	[64]	Retrospective	31 patients/32 tumors	Grade 2:2	52	16 M 15 F	PET	TBRmax	2.61	80%	86%	0.78	81%
					Grade 3:3				TBRmean §	-	-	-	0.74	-
					Grade 4:27				TTP §	-	-	-	0.71	-
									Slope §	-	-	-	0.70	-
								PWI rCBF	TBRmax §	-	-	-	0.65	-
									TBRmean §	-	-	-	0.55	-
								PWI rCBV	TBRmax §	-	-	-	0.58	-
									TBRmean §	-	-	-	0.64	-
								PWI MTT	TBRmax §	-	-	-	0.59	-
									TBRmean §	-	-	-	0.59	-
	Pyka et al., 2018	[65]	Retrospective	47 patients/63 lesions	Grade 2:5	54	22 M 25 F	PET	TBR30–40 min	2.07	80%	85%	0.863	
					Grade 3:20				TBR10–20 min	1.71	76%	85%	0.848	
					Grade 4:38				TTP	20 min	64%	79%	0.728	
								PWI MRI	rCBVuncor	4.32	62%	77%	0.726	
									rCBVcor	3.35	66%	77%	0.708	
								DWI MRI	ADC	1610 × 10^−6^ mm^2^/s	50%	77%	0.688	
									nADC	1.22	62%	77%	0.697	
									FA §	98.9	65%	62%	0.593	
								PET/MRI	TBR30–40 min + TTP + rCBVcor + nADC	-	78%	92%	0.891	
	Werner et al., 2021	[66]	Retrospective	23	Grade 4:23	58	13 M 10 F	PET	TBRmax	2.85	64%	92%	0.75	78%
									TBRmean	1.95	82%	92%	0.77	87%
									Slope §	0.02 SUV/h	73%	75%	0.72	74%
									TTP	35 min	64%	83%	0.82	74%
									TBRmax + TTP	2.85 and 35 min	36%	100%	-	70%
									TBRmean + TTP	1.95 and 35 min	55%	100%	-	78%
								MRI	RANO criteria §	-	30%	79%	-	58%
	Galldiks et al., 2015	[67]	Retrospective	22	Grade 4:22	56	14 M 8 F	PET	TBRmax	2.3	100%	91%	0.94	96%
									TBRmean	2.0	82%	82%	0.91	82%
									Kinetic pattern	II/III	-	-	-	-
									TBRmax+ Kinetic pattern	2.3 and II/III	80%	91%	-	86%
									TBRmean+ Kinetic pattern	2.0 and II/III	60%	91%	-	76%
	Werner et al., 2019	[68]	Retrospective	48	Grade 3:8	50	29 M 19 F	PET	TBRmax	1.95	100%	79%	0.89	83%
					Grade 4:40				TBRmean	1.95	100%	79%	0.89	83%
									TTP	32.5 min	80%	69%	0.79	72%
									Slope	0.32 SUV/h	70%	75%	0.82	74%
									TBRmax/mean + TTP	1.95 and 32.5 min	89%	91%	-	90%
									TBRmax/mean + Slope	1.95 and 0.32 SUV/h	78%	97%	-	93%
								DWI-MRI	Visual assessment §	-	70%	66%	-	67%
									ADC §	1.09×10^−3^ mm^2^/s	60%	71%	0.73	69%
								PET/MRI	TBRmax/mean + ADC	-	67%	94%	-	89%
	Lohmann et al., 2020	[69]	Retrospective	34	Grade 3:1	57	21 M 13 F	PET	TBRmax	2.25	81%	67%	0.79	74%
					Grade 4:33				TBRmean	1.95	75%	61%	0.73	68%
									TTP §	25 min	75%	44%	0.61	59%
									Slope §	0.3 SUV/h	56%	61%	0.55	59%
									TBRmean + TBRmax	-	75%	72%	-	74%
									TBRmean + TTP	-	69%	78%	-	74%
									TBRmean + Slope §	-	50%	78%	-	65%
									TBRmax + TTP	-	69%	83%	-	76%
									TBRmax + Slope	-	50%	89%	-	71%
									TTP + Slope §	-	56%	61%	-	59%
									TBRmax + TBRmean + TTP	-	69%	89%	-	79%
									Radiomics features	-	100%	40%	0.74	70%
	Kebir et al., 2016	[70]	Retrospective	26	Grade 4:26	58	21 M 5 F	PET	TBRmax	1.9	84%	86%	0.88	85%
									TBRmean	1.9	74%	86%	0.86	77%
									TAC	II/III	84%	100%	-	89%
									TTP	-	-	-	0.86	-
	Rachinger et al., 2005	[71]	Retrospective	45	Grade 1:1	45	23 M 22 F	PET	SUVmax	2.2	100%	93%		
					Grade 2:10			MRI	Volume/Gd-enhancing area	∆25%/new area	94%	50%		
					Grade 3:12									
					Grade 4:22									
	Lohmeier et al., 2019	[72]	Retrospective	42	Grade 1–2:2	47	32 M 10 F	PET	SUVmax §	-	-	-	-	
					Grade 3–4:40				SUV80mean §	-	-	-	-	
									SUV-BG §	-	-	-	-	
									TBR80mean	-	-	-	-	
									TBRmax	2.0	81%	60%	0.81	
								DWI-MRI	ADCmean	1254 × 10^−6^ mm^2^/s	62%	100%	0.82	
									ADC-BG §	-	-	-	-	
									rADCmean	-	-	-	-	
								PET/MRI	TBRmax + ADCmean	-	97%	60%	0.90	
	Bashir et al., 2019	[73]	Retrospective	146	Grade 4:146	60	96 M 50 F	PET	TBRmax	2.0	99%	94%	0.970	99%
									TBRmean	1.8	96%	94%	0.977	96%
									BTV	0.55 cm^3^	98%	94%	0.955	98%
	Steidl et al., 2020	[74]	Retrospective	104	Grade 2:9	52	68 M 36 F	PET	TBRmax	1.95	70%	60%	0.72	68%
					Grade 3:24				TBRmean	-	-	-	0.72	-
					Grade 4:71				TTP §	-	-	-	0.60	-
									Slope	0.69 SUV/h	84%	62%	0.69	80%
									TBRmax + Slope #	1.95 and/or 0.69 SUV/h	96%	43%	-	86%
								MRI	rCBVmax	2.85	54%	100%	0.75	63%
								PET/MRI	rCBVmax + TBRmax + Slope #	-	98%	43%	-	87%
	Pöpperl et al., 2006	[75]	Prospective	24	Grade 3:5	49	15 M 9 F	PET	Tumax/BG #	2.0	100%	78%		
					Grade 4:19				Tumax/BG #	2.1	97%	91%		
									Tumax/BG #	2.2	82%	95%		
									Tumax/BG #	2.3	74%	98%		
									Tumax/BG #	2.4	74%	100%		
									Tumax/BG #	2.5	62%	100%		
									Visual analysis #	Nodular vs. non-nodular	94%	94%		
	Müller et al., 2022	[76]	Retrospective	151	Grade 2:28	52	97 M 54 F	PET	TBRmax	-	-	-	-	
					Grade 3:40				TBRmean	-	-	-	-	
					Grade 4:83				TBRmax + TBRmean #	-	66%	80%	0.78	
									Radiomics features #	-	73%	80%	0.85	
									TBRmax + TBRmean + radiomics features #	-	81%	70%	0.85	
	Mehrkens et al., 2008	[77]	Prospective	31	Grade 2:17	46	17 M 14 F	PET	SUVmax/BG §	2.0				
					Grade 3:6									
					Grade 4:8									
	Galldiks et al., 2015	[78]	Retrospective	124	Grade 2:55	52	81 M 43 F	PET	TBRmax	2.3	68%	100%	0.85	71%
					Grade 3:19				TBRmean	2.0	74%	91%	0.91	75%
					Grade 4:50				TTP	45 min	82%	73%	0.81	81%
									Curve pattern	II/III	78%	73%	-	77%
									TBRmax + Curve pattern	2.3 and/or II/III	93%	73%	-	91%
									TBRmean + Curve pattern	2.0 and/or II/III	93%	73%	-	91%
									TBRmax + TTP	2.3 and/or 45 min	92%	73%	-	90%
									TBRmean + TTP	2.0 and/or 45 min	93%	100%	-	93%
								MRI	RANO criteria §	-	92%	9%	-	85%
	Pöpperl et al., 2004	[79]	Prospective	53	Grade 1:1	-	28 M 25 F	PET	SUVmax	2.2				
					Grade 2:9				SUVmax/BG	2.0				
					Grade 3:16				SUV80/BG	-				
					Grade 4:27				SUV70/BG	-				
Prognosis/Treatment response evaluation														
	Müther et al., 2019	[80]	Prospective	31	Grade 4:31	67	13 M 18 F	PET	Volume	4.3 cm^3^				
	Jansen et al., 2013	[61]	Prospective	33	Grade 3:20	-	-	PET	Uptake kinetics	Increasing				
					Grade 4:13									
	Suchorska et al., 2018	[81]	Retrospective	61	Grade 2:44	46	31 M 30 F	PET	Initial BTV §	-				
					Grade 3:17				Initial TBRmax §	-				
									Initial TAC §	Increasing vs. decreasing				
									BTV after 6 months	-				
									TBRmax after 6 months §	-				
									TAC after 6 months §	Increasing vs. decreasing				
									BTV response	∆ ± 25%				
									TBRmax response	∆ ± 10%				
									TAC response §	Stable increasing vs. Decreasing to increasing vs. Increasing to decreasing vs. Stable decreasing				
									FET-PET response	Yes vs. no				
								MRI	Initial T2 volume	-				
									T2 volume after 6 months	-				
									T2 volume response §	RD vs. SD vs. PD				
	Galldiks et al., 2012	[82]	Prospective	25	Grade 4:25	54	15 M 10 F	PET	TBRmax change	∆-10% (PFS)/∆-20% (OS)	83% (OS)	67% (OS)	0.75 (OS)	
									TBRmean change	∆-5%	67%	75%	0.72	
									Tvol 1.6 change	∆0% (PFS)	-	-	-	
								MRI	Gd-volume §	∆0%/∆-25%	-	-	-	
	Suchorska et al., 2015	[83]	Prospective	79	Grade 4:79	-	-	PET	BTVpreRCx	9.5 cm^3^	64%	70%		
									LBRmax-preRCx	2.9 (OS)	68%	73%		
									Initial TAC	Increasing vs. decreasing (OS)	-	-		
								MRI	Gd+ volume	-	-	-		
	Jansen et al., 2014	[84]	Retrospective	59	Grade 2:59	43	32 M 27 F	PET	TAC	Increasing vs. decreasing				
									Uptake §	Positive vs. negative				
									SUVmax/BG §	-				
									SUVmean/BG §	-				
									SUVtotal/BG §	-				
									BTV §	-				
								MRI	Contrast enhancement §	Yes vs. no				
									Largest diameter	6 cm (PFS)				
									Tumor crossing midline §	Yes vs. no				
	Thon et al., 2015	[85]	Prospective	98	Grade 2:54	-	56 M 42 F	PET	TAC	Homogeneous decreasing vs. focal decreasing vs. homogeneous increasing				
					Grade 3:40				SUVmax §	2.3				
					Grade 4:4			MRI	Tumor volume §	35 mL				
	Kunz et al., 2018	[86]	Prospective	98	Grade 2:59	-	-	PET	TAC	Homogeneous increasing vs. mixed vs. homogeneous decreasing				
					Grade 3:35				TTPmin	>25 min vs. 12.5 < t ≤ 25 min vs. ≤12.5 min				
					Grade 4:4				SUVmax §	2.3				
								MRI	Tumor volume §	35 mL				
	Ceccon et al., 2021	[87]	Prospective	41	Grade 2:1	52	22 M 19 F	PET	TBRmax baseline	2.0 (PFS)/1.9 § (OS)				
					Grade 3:2				TBRmean baseline §	1.9 (PFS)/1.8 (OS)				
					Grade 4:38				MTV baseline	28.2 mL (PFS)/13.8 mL (OS)				
									TBRmax change	0%				
									TBRmean change §	0%				
									MTV change	0%				
								MRI	RANO criteria §	SD/PR/CR vs. PD				
	Galldiks et al., 2018	[88]	Prospective	21	Grade 4:21	55	13 M 8 F	PET	TBRmax relative reduction §	27%	92%	63%	0.78	
									TBRmean relative reduction §	16%	92%	63%	0.81	
									MTV relative reduction §	27%	77%	63%	0.82	
									Absolute MTV at follow-up	5 mL	85%	88%	0.92	
								MRI	RANO criteria §	PR or SD	63%	69%	-	
	Carles et al., 2021	[89]	Prospective	32	Grade 4:32	52	17 M 15 F	PET	Radiomic features:					
									SUVmin &	-				
									SUVmean &	-				
									GLV &	-				
									GLV2 &	-				
									WF_GLV &	-				
									Qacor &	-				
									QHGZE &	-				
									QSZHGE &	-				
									QGLN2 &	-				
									QHGRE &	-				
									QSRHGE &	-				
									QLRHGE &	-				
									SZLGE	-				
									Busyness &	-				
									WF_TS &	-				
									QvarianceCM &	-				
									Eccentricity &	-				
									SUVmean + WF_GLV + QLRHGE + SUVmin	-				
									SZLGE + Busyness + QVarianceCM + Eccentricity	-				
	Suchorska et al., 2018	[90]	Retrospective	300	Grade 2:121	48	166 M 134 F	PET	TBRmax §	1.6				
					Grade 3:106				TBRmax §	2.6				
					Grade 4:73				TTPmin	17.5 min (OS)				
								MRI	Contrast enhancement §	Yes vs. no				
									T2 volume §	49 mL				
	Wirsching et al., 2021	[91]	Retrospective	31	Grade 4:31	-	-	PET	TBR in non-contrast enhancing tumor portions at follow-up	High vs. low				
								MRI	Contrast enhancement at baseline	-				
									ADC at baseline	-				
									Contrast enhancement at follow-up	-				
	Sweeney et al., 2013	[92]	Retrospective	28	Grade 2:5	-	21 M 7 F	PET	SUVmax	2.6				
					Grade 3:12				TBRmax §	-				
					Grade 4:11				TBRmean§	-				
									Tumor volume §					
									VolSUVmax ≥ 2.2	-				
									Vol ≥ 40%SUVmax	-				
								MRI	VolMRI	-				
								PET/MRI	VolMRI + VolSUVmax ≥ 2.2	-				
									VolMRI + Vol≥ 40%SUVmax	-				
									Non-overlap, VolMRI + VolSUVmax ≥ 2.2	-				
									Non-overlap, VolMRI + Vol ≥ 40%SUVmax	-				
	Pyka et al., 2014	[93]	Retrospective	34	Grade 1:2	41	22 M 12 F	PET	TBRmax	2.5			0.696	
					Grade 2:19				TBRmean	2.3			0.696	
					Grade 3:3				TTP	20 min			0.848	
					Grade 4:10				Peak TBR	2.2			0.704	
									Slope-to-peak	7 × 10^−5^/s			0.711	
	Wollring et al., 2022	[94]	Retrospective	36	Grade 3:8	54	20 M 16 F	PET	New distant FET hotspot	Yes vs. no				
					Grade 4:28				TBRmax change	0%				
									TBRmean change §	0%				
									MTV change	0%				
									TTP change §	0%				
								MRI	RANO criteria	SD/PR/CR vs. PD				
	Bauer et al., 2020	[95]	Retrospective	60	Grade 3:15	55	35 M 25 F	PET	TBRmax §	2.55	70%	57%	0.63	
					Grade 4:45				TBRmean §	2.05	60%	70%	0.69	
									MTV §	11.15 mL	72%	54%	0.56	
									TTP	25 min	90%	87%	0.90	
									Slope §	−0.103 SUV/h	70%	90%	0.77	
	Piroth et al., 2011	[96]	Prospective	44	Grade 4:44	57	16 M 28 F	PET	VolTBR ≥ 1.6	25 mL				
									VolTBR ≥ 2.0	10 mL				
									TBRmax	2.4				
									TBRmean	2.0				
								MRI	Gd-volume §	10 mL				
	Jansen et al., 2015	[97]	Retrospective	121	Grade 3:51	54	73 M 48 F	PET	TTPmin	12.5 min				
					Grade 4:70				SUVmax/BG §	-				
									SUVmean/BG §	-				
									BTV §	-				
								MRI	contrast enhancement §	Yes vs. no				
	Moller et al., 2016	[98]	Prospective	31	Grade 3:6	54	-	PET	BTV baseline	-				
					Grade 4:25				Tmax/B baseline #	-				
									∆BTV scan 2 §	-				
									∆BTV scan 3 §	-				
									∆Tmax/B scan 2 #	-				
									∆Tmax/B scan 3 #	-				
								MRI	Volume (+necrosis) §	-				
									Volume (−necrosis)	-				
	Dissaux et al., 2020	[99]	Prospective	29	Grade 3:3	60	17 M 12 F	PET	TBRmax	Median (5.03)				
					Grade 4:26				TBRmean §	Median				
									SUVmax §	Median				
									SUVmean §	Median				
									SUVpeak §	Median				
									TLG §	Median				
									Volume §	Median				
	Piroth et al., 2011	[100]	Prospective	22	Grade 4:22	56	13 M 9 F	PET	Volume	20 mL				
									TBRmax §	3.0				
									TBRmean §	2.0				
									TBRmean	2.4				
									Early TBRmax response	∆-10%				
									Early TBRmean response	∆-10%				
								MRI	Diameter of contrast-enhanced area	4 cm				
	Schneider et al., 2020	[101]	Retrospective	42	Grade 2:19	46	26 M 16 F	PET	SUVmax	3.4				
					Grade 3:23				TBRmax	3.03				
									BTV	10 cm^3^				
	Kertels et al., 2019	[102]	Retrospective	35	Grade 2:14	48	20 M 15 F	PET	FET positivity	Yes vs. no				
					Grade 3:21									
	Floeth et al., 2007	[103]	Prospective	33	Grade 2:33	-	20 M 13 F	PET	Mean FET uptake	1.1				
									Maximum FET uptake §	2.0				
								MRI	Hemisphere§	Right vs. left				
									Brain lobe location §	-				
									Extension §	Deep vs. superficial				
									Size §	3 cm				
									Mass shift §	Yes vs. no				
									Appearance	Circumscribed vs. diffuse				
								PET/MRI	Mean FET uptake + MRI appearance	-				
	Niyazi et al., 2012	[104]	Retrospective	56	Grade 3:13	50	34 M 22 F	PET	Kinetics pre re-RT	G1–2 vs. G3 vs. G4–5				
					Grade 4:43				Kinetics post re-RT §	G1–2 vs. G3 vs. G4–5				
									SUVmax/BG pre re-RT §	3.3				
									SUVmax/BG post re-RT §	2.6				
									SUVmean/BG pre re-RT §	2.2				
									SUVmean/BG post re-RT §	2.3				
									BTV pre re-RT §	13.7 cc				
									BTV post re-RT §	7.3 cc				
	Pyka et al., 2016	[39]	Retrospective	113	Grade 3:26	59	43 M 70 F	PET	TBRmax §	2.5				
					Grade 4:87				TBRmean §	1.56 (PFS)/1.57 (OS)				
									MTV	19.4 (PFS) §/18.9 (OS)				
									TLU	35.0 (PFS) §/17.1 (OS)				
									Textural parameters:					
									Coarseness	5.96 × 10^−3^ (PFS)/6.88 × 10^−3^ (OS)				
									Contrast	0.427				
									Busyness	1.366 (PFS)/0.984 (OS)				
									Complexity	0.085 (PFS)/0.094 (OS)				
	Blanc-Durand et al., 2018	[43]	Retrospective	37	Grade 1:3	45	23 M 14 F	PET	TBRmax §	-				
					Grade 2:15				TBRmean §	-				
					Grade 3:14				TTP	-				
					Grade 4:5				Slope	-				
									TAC	-				

**Table 3 pharmaceuticals-17-01228-t003:** Different tumor-to-brain ratios and the methodology used to obtain them.

Parameter	Definition
**TBR_mean_**	Mean uptake in the tumor area with a TBR ≥ 1.6 divided by mean uptake in the normal brain
**TBR_max_**	Maximal uptake in the tumor area divided by mean uptake in the normal brain
**TBR_10/16mm_**	Mean uptake in a ROI/VOI with a diameter of 10/16 mm centered on the tumor area with the highest uptake divided by mean uptake in the normal brain
**TBR_25mm2_**	Mean uptake in a standardized ROI/VOI with a size of 25 mm^2^ placed manually at the biopsy sites centered to the titanium pellets on postoperative images divided by mean uptake in the normal brain
**TBR_3SD_**	Mean uptake in an isocontour region around the lesion maximum using a cutoff of three standard deviations above average activity in the reference region divided by mean uptake in the normal brain
**TBR_70/80%_**	Mean in a 70/80% isocontour region divided by mean uptake in the normal brain
**TBR**	Uptake in the tumor area (unspecified) divided by mean uptake in the normal brain
**SUV_max/mean_/BG**	SUV_max/mean_ of the tumor area divided by maximal uptake in the normal brain

**Table 4 pharmaceuticals-17-01228-t004:** Summary of PET parameters. *: reached significance, X: did not reach significance, &: did not stay significant after Bonferroni multiple-test correction, NA: not available. TBR_max_: L_max_/B, SUV_max_/BG, LNR, TNR, LBR_max_, T/Wm, TBR_max(20–40min)_, T_max_/B, maximum FET uptake, Tu_max_/BG; TBR_3SD_: L_mean_/B, mean FET uptake; TBR_25mm2_: TBR, FET ratio; TBR_10mm_: TBR_mean_; TBR_16mm_: TBR_mean_, TBR_max_; TBR_70%_: SUV_70_/BG; TBR_80%_: SUV_80_/BG; TBR: UR, FET lesion/brain ratio, FET uptake, tumor/brain tissue ratio, TBR_mean_, TBR_max_; TAC: kinetic pattern, curve pattern; TTP: Tpeak; BTV: volume, MTV, Vol, T_vol 1.6_; radiomic features: textural parameters.

Indication	Number of Studies	Grade	Parameters	Threshold	Sensitivity	Specificity	AUC	Accuracy	Significance
Diagnosis									
	1	LGG and HGG	Visual grading system	-	-	-	-	-	NA
	1	LGG and HGG	TBR_max_	-	-	-			NA
	1	LGG and HGG	TBR_25mm2_	1.6	92%	81%		-	*
	1	LGG and HGG	TBR_3SD_	-	-	-			NA
	1	LGG and HGG	TBR	1.6	88%	88%		-	*
	1	LGG and HGG	^18^F-FET_n_ uptake	1.4 x background	76%	80%	0.89	78%	*
Grading (LGG vs. HGG)									
	1	LGG and HGG	FET uptake	Reduced vs. normal vs. increased	-	-			NA
	1	LGG and HGG	FET uptake pattern	Inhomogeneous vs. diffuse vs. focal	-	-			X
	1	LGG and HGG	Early SUV	2.32	73%	71%		72%	*
	1	LGG and HGG	Middle SUV	-	-	-	-	-	X
	1	LGG and HGG	Late SUV	-	-	-	-	-	X
	1	LGG and HGG	e-m Ratio	0.93	93%	94%		94%	*
	1	LGG and HGG	e-l Ratio	0.95	87%	88%		87%	*
	1	LGG and HGG	SoD	0.5	93%	82%		87%	*
	1	LGG and HGG	SUV_max_	-	-	-			*
Grade 2/3 vs. Grade 4	1	LGG and HGG	SUV_sd_	0.45	67%	87%	0.816	83%	*
Grade 2/3 vs. Grade 4	1	LGG and HGG	SUV_max_/BG	-	-	-			*
	2	LGG and HGG	SUV_mean_/BG	-	-	-			X
Grade 2 vs. 3		LGG and HGG		-	-	-			X
Grade 2 vs. 3	1	LGG and HGG	SUV_total_/BG	-	-	-			X
	1	LGG and HGG	SUV_90_ 10–60 min	0.2	94%	100%	0.969		*
	1	LGG and HGG	SUV_90_ 15–60 min	−0.41	94%	100%	0.965		*
	1	LGG and HGG	TBR_max(0–10min)_	2.8	76%	79%		76%	*
	1	LGG and HGG	TBR_max(5–15min)_	2.7	78%	76%		77%	*
	1	LGG and HGG	TBR_max(5–20min)_	2.6	80%	74%		76%	*
	1	LGG and HGG	TBR_max(10–30min)_	2.5	75%	75%		74%	*
	7	LGG and HGG	TBR_max_	2.58	71%	85%	0.798		*
		LGG and HGG		2.62	82%	68%	0.83	78%	*
Grade 2/3 vs. Grade 4		LGG and HGG		2.67	92%	61%	0.824	67%	*
		LGG and HGG		2.7	67%	78%		70%	*
		LGG and HGG		-	-	-			*
		LGG and HGG		-	-	-			X
Grade 2 vs. 3		LGG and HGG		-	-	-			X
Grade 2/3 vs. Grade 4	1	LGG and HGG	TBR_peak_	2.35	92%	61%	0.832	67%	*
	2	LGG and HGG	TBR_mean_	2	83%	58%	0.65	75%	X
Grade 2/3 vs. Grade 4		LGG and HGG		2.31	58%	93%	0.791	86%	*
	1	LGG and HGG	∆TBR_mean_ 20–40 min/70–90 min	−8%	83%	75%	0.85	81%	*
	1	LGG and HGG	TBR_16mm_	1.69	82%	68%	0.8	78%	*
Grade 3 vs. 4	3	HGG	TBR	1.68	-	-	0.644		*
Grade 3 vs. 4		HGG		2.74	-	-	0.614		X
		LGG and HGG		3	-	-			*
	4	LGG and HGG	TTP	25 min	87%	100%		94%	*
		LGG and HGG		30 min	54%	91%	0.78	65%	*
		LGG and HGG		35 min	58%	92%	0.76	69%	*
Grade 2/3 vs. Grade 4		LGG and HGG		-	-	-	-		X
	1	LGG and HGG	Slope	−0.03 SUV/h	64%	91%	0.78	72%	*
	7	LGG and HGG	TAC	II/III	88%	75%		83%	*
		LGG and HGG		I/II vs. III	73%	100%		87%	NA
		LGG and HGG		Decreasing	90%	66%		80%	NA
Grade 2 vs. 3		LGG and HGG			88%	63%			NA
		LGG and HGG		Increasing vs. Decreasing	95%	72%			NA
		LGG and HGG			96%	94%			*
Grade 2/3 vs. Grade 4		LGG and HGG		LGG-like vs. mixed vs. HGG-like	-	-	-		NA
Grade 2/3 vs. Grade 4	1	LGG and HGG	COV	27.21	58%	91%	0.808	84%	*
Grade 2/3 vs. Grade 4	1	LGG and HGG	HI	1.77	67%	87%	0.826	83%	*
Grade 3 vs. 4	4	HGG	BTV	19.7	-	-	0.71		*
Grade 2/3 vs. Grade 4		LGG and HGG		20.13	75%	80%	0.801	79%	*
		LGG and HGG		-	-	-			X
Grade 2 vs. 3		LGG and HGG		-	-	-			X
Grade 3 vs. 4	2	HGG	TLU	46.2	-	-	0.704		*
Grade 2/3 vs. Grade 4		LGG and HGG		50.93	75%	83%	0.841	81%	*
Grade 2/3 vs. Grade 4	1	LGG and HGG	Relative K1	-	85%	60%	0.766		*
Grade 2/3 vs. Grade 4	1	LGG and HGG	Relative K2	-	-	-	-		X
Grade 2/3 vs. Grade 4	1	LGG and HGG	Relative K3	-	-	-	-		X
Grade 2/3 vs. Grade 4	1	LGG and HGG	Relative FD	-	67%	78%	0.716		*
Grade 2/3 vs. Grade 4	1	LGG and HGG	TBR_max_ + SUV_sd_ + TBR_mean_	-	75%	85%	0.850	83%	*
Grade 2/3 vs. Grade 4	1	LGG and HGG	HI + SUV_sd_ + MTV	-	75%	83%	0.848	81%	*
Grade 2/3 vs. Grade 4	1	LGG and HGG	HI + SUV_sd_ + TLU	-	75%	84%	0.848	81%	*
Grade 2/3 vs. Grade 4	1	LGG and HGG	SUV_max_/BG + TTP	-	-	-	0.745		*
Grade 2/3 vs. Grade 4	1	LGG and HGG	SUV_max_/BG + TTP + relative K1 + relative FD	-	-	-	0.799		*
	1	LGG and HGG	Logistic regression using early SUV + SoD	50%	93%	100%		97%	X
			Radiomic features:						*
Grade 3 vs. 4	1	HGG	Coarseness	0.607	-	-	0.757		*
Grade 3 vs. 4	1	HGG	Contrast	0.203	-	-	0.775		*
Grade 3 vs. 4	1	HGG	Busyness	1.12	-	-	0.737		*
Grade 3 vs. 4	1	HGG	Complexity	0.069	-	-	0.633		*
Grade 3 vs. 4	1	HGG	Combined	2.05	-	-	0.830		*
IDH status determination									
	2	LGG and HGG	SUV_sd_	0.11	47%	57%	0.710	66%	*
		LGG and HGG		0.23	-	-	-	-	*
	5	LGG and HGG	TBR_max_	2.07	8%	100%	0.59	71%	X
		LGG and HGG		2.21	48%	87%	0.658	72%	*
		LGG		-	-	-	-	-	X
		LGG and HGG		-	-	-	-	-	X
		LGG and HGG		-	-	-	-	-	*
	2	LGG and HGG	TBR_peak_	2.15	57%	73%	0.638	67%	X
		LGG and HGG		-	-	-	-	-	X
	5	LGG and HGG	TBR_mean_	1.68	12%	100%	0.66	73%	*
		LGG and HGG		1.84	62%	68%	0.633	66%	X
		LGG and HGG		1.85	44%	92%	0.73	69%	*
		LGG and HGG		-	-	-	-	-	X
		LGG and HGG		-	-	-	-	-	*
	1	LGG and HGG	TBR_16mm_	2.15	56%	77%	0.68	67%	*
	3	LGG	TBR	1.3	89%	36%	-	-	NA
		LGG		1.6	71%	53%	-	-	NA
		LGG		2.0	57%	68%	-	-	NA
	3	LGG and HGG	TTP	25 min	86%	60%	0.75	72%	*
		LGG and HGG		45 min	27%	93%	0.75	73%	*
		LGG and HGG		-	-	-	-	-	*
	3	LGG and HGG	Slope	−0.26 SUV/h	81%	60%	0.75	70%	*
		LGG and HGG		0.30 SUV/h	58%	90%	0.79	80%	*
		LGG and HGG		-	-	-	-	-	*
	1	LGG and HGG	TAC	centroid #1 vs. centroid #3	-	-	-	-	*
	1	LGG and HGG	COV	8.85	52%	76%	0.65	67%	*
	1	LGG and HGG	HI	1.26	48%	87%	0.676	72%	*
	2	LGG and HGG	BTV	19.48	90%	46%	0.66	62%	*
		LGG and HGG		-	-	-	-	-	X
	2	LGG and HGG	TLU	28.95	81%	57%	0.698	66%	*
		LGG and HGG		-	-	-	-	-	X
	1	LGG and HGG	TBR_mean_ + TBR_16mm_	1.85 and 2.15	44%	91%	-	69%	*
	1	LGG and HGG	TTP + Slope	25 min and −0.26 SUV/h	77%	70%	-	73%	*
	1	LGG and HGG	TBR_mean_ + TTP	1.85 and 25 min	40%	96%	-	69%	*
	1	LGG and HGG	TBR_16mm_ + TTP	2.15 and 25 min	51%	94%	-	73%	*
	1	LGG and HGG	TBR_mean_ + Slope	1.85 and −0.26 SUV/h	40%	94%	-	68%	*
	1	LGG and HGG	TBR_16mm_ + Slope	2.15 and −0.26 SUV/h	47%	91%	-	70%	*
	1	LGG and HGG	TBR_max_ + SUV_sd_ + TBR_mean_	-	76%	84%	0.821	81%	*
	1	LGG and HGG	HI + SUV_sd_ + MTV	-	86%	81%	0.804	83%	*
	1	LGG and HGG	HI + SUV_sd_ + TLU	-	76%	84%	0.799	81%	*
	1	LGG and HGG	Midline involvement	Yes vs. no	-	-	-	-	*
	1	LGG and HGG	Simple predictive model	-	85%	71%	0.786	76%	*
	1	LGG and HGG	PET-Radiomics model	-	80%	74%	0.812	76%	*
	1	LGG and HGG	Slope + Radiomic feature SZHGE	-	54%	93%	-	81%	*
			Radiomic features:						*
	1	LGG and HGG	SkewnessH	-	31%	90%	0.53	71%	*
	1	LGG and HGG	LRHGE	-	8%	100%	0.52	71%	*
Prediction of oligodendroglial components									
	1	LGG and HGG	SUV_mean_/BG	2.1	61%	59%			*
	1	LGG and HGG	SUV_total_/BG	6.9	75%	66%			*
	2	LGG and HGG	TBR_max_	2.6	70%	72%			*
		LGG		-	-	-			*
	3	LGG	TBR	1.3	100%	23%			NA
		LGG		1.6	93%	48%			NA
		LGG		2	86%	65%			NA
	1	LGG and HGG	BTV	4 mL	71%	69%			*
Guided resection/biopsy									
	1	HGG	BTV	1 cm^3^					*
	1	LGG and HGG	TBR_25mm2_	1.6	-	-			*
	3	LGG	TBR	1.6	54%	12%			*
		HGG			88%	46%			*
		LGG and HGG		-	-	-	0.76		*
Detection of residual tumor									
	1	HGG	TBR	1.6	-	-			*
	1	LGG and HGG	Visual uptake	>Background	-	-			*
Guided radiotherapy									
	7	HGG	SUV_max_	30%	-	-			NA
		HGG		40%	-	-			NA
		HGG		50%	-	-			NA
		HGG		60%	-	-			NA
		HGG		70%	-	-			NA
		HGG		80%	-	-			NA
		HGG		90%	-	-			NA
	1	HGG	TBR_max_	1.6	-	-			NA
	5	HGG	TBR	1.6	-	-			NA
		HGG			-	-			NA
		HGG			-	-			NA
		HGG			-	-			NA
		HGG			-	-			NA
Detection of malignant transformation in LGG									
	3	LGG	TBR_max_	∆ + 33%	72%	89%	0.87	78%	*
		LGG and HGG		2.46	82%	89%	0.92	85%	*
		LGG		-	57%	41%	0.476		X
	1	LGG	TBR_mean_	∆ + 13%	72%	78%	0.8	74%	*
	2	LGG	TTP	∆-6 min	72%	89%	0.78	78%	*
		LGG		25 min	57%	47%	0.511		X
	1	LGG and HGG	TTP_min_	17.5 min	73%	67%	-	70%	*
	1	LGG	TAC	-	71%	41%	0.549		X
	1	LGG	TAC change	I to II/III	72%	89%	-	78%	*
	1	LGG	TBR_max_ + TTP + TAC change	∆ + 33% or ∆-6 min or I to II/III	83%	78%	-	81%	*
	1	LGG	TBR_max_ + TAC + TTP	1.6 + II/III + 25 min	65%	58%	0.634		X
	1	LGG	TBR_max_ + TAC	1.6 + II/III	65%	58%	0.639		X
	1	LGG	TBR_max_ + TTP	1.6 + 25 min	96%	25%	0.591		X
Recurrence vs. treatment-related changes									
	1	HGG	Visual analysis	Nodular vs. non-nodular	94%	94%			NA
	6	LGG	SUV_max_	1.48	88%	89%	0.951		*
		LGG and HGG		1.66	87%	100%	0.978		*
		HGG			93%	100%	0.993		*
		LGG and HGG		2.2	100%	93%			*
		LGG and HGG			-	-			*
		LGG and HGG		-	-	-			X
	1	LGG and HGG	SUV80_mean_	-	-	-			X
	1	LGG and HGG	SUV-BG	-	-	-			X
	20	LGG	TBR_max_	1.64	100%	75%	0.893		*
		LGG and HGG		2	81%	60%	0.81		*
		LGG and HGG			-	-			X
		LGG and HGG			-	-			*
		HGG			99%	94%	0.970	99%	*
		HGG			100%	78%			NA
		HGG		2.1	97%	91%			NA
		LGG and HGG		2.18	86%	88%	0.940		*
		HGG		2.2	82%	95%			NA
		HGG		2.3	74%	98%			NA
		HGG		2.4	74%	100%			NA
		HGG		2.46	86%	100%	0.964		*
		HGG		2.5	62%	100%			NA
		LGG and HGG		2.61	80%	86%	0.78	81%	*
		HGG		2.65	80%	88%			*
		HGG		2.85	64%	92%	0.75	78%	*
		HGG		3.44	86%	88%	0.86		*
		HGG		3.58	64%	100%	0.84		*
		HGG		3.69	79%	88%	0.86		*
		LGG and HGG		-	-	-	-		*
	1	HGG	TBR_max_ after 6 months	-	-	-			*
	11	HGG	TBR_mean_	1.8	96%	94%	0.977	96%	*
		HGG		1.9	74%	86%	0.86	77%	*
		HGG		1.95	82%	92%	0.77	87%	*
		HGG			100%	79%	0.89	83%	*
		HGG			75%	61%	0.73	68%	*
		LGG and HGG		2.0	74%	91%	0.91	75%	*
		HGG			82%	82%	0.91	82%	*
		HGG		2.19	71%	88%	0.80		*
		HGG		2.31	61%	100%	0.83		*
		LGG and HGG		-	-	-	0.72		*
		LGG and HGG		-	-	-	-		*
	1	LGG and HGG	TBR_30–40min_	2.07	80%	85%	0.863		*
	1	LGG and HGG	TBR_10–20min_	1.71	76%	85%	0.848		*
	1	HGG	TBR_10mm_	2.86	86%	75%	0.81		*
	8	HGG	TBR_16mm_	1.9	84%	86%	0.88	85%	*
		LGG and HGG		1.95	70%	60%	0.72	68%	*
		HGG			100%	79%	0.89	83%	*
		HGG		2.25	81%	67%	0.79	74%	*
		LGG and HGG		2.3	68%	100%	0.85	71%	*
		HGG			100%	91%	0.94	96%	*
		HGG		2.44	82%	75%	0.82		*
		LGG and HGG		-	-	-	0.74		X
	2	HGG	TBR_70%_	2.72	86%	88%	0.87		*
		LGG and HGG		-	-	-			*
	2	HGG	TBR_80%_	3.08	82%	88%	0.88		*
		LGG and HGG		-	-	-			*
	1	HGG	TBR_90%_	3.23	71%	100%	0.85		*
	1	LGG and HGG	TBR80_mean_	-	-	-			*
	8	LGG and HGG	TTP	20 min	64%	79%	0.728		*
		HGG		25 min	75%	44%	0.61	59%	X
		HGG		32.5 min	80%	69%	0.79	72%	*
		HGG		35 min	64%	83%	0.82	74%	*
		LGG and HGG		45 min	82%	73%	0.81	81%	*
		LGG and HGG		-	-	-	0.60		X
		LGG and HGG		-	-	-	0.71		*
		HGG		-	-	-	0.86	-	*
	5	HGG	Slope	0.02 SUV/h	73%	75%	0.72	74%	X
		HGG		0.3 SUV/h	56%	61%	0.55	59%	X
		HGG		0.32 SUV/h	70%	75%	0.82	74%	*
		LGG and HGG		0.69 SUV/h	84%	62%	0.69	80%	*
		LGG and HGG		-	-	-	0.70		*
	3	LGG and HGG	TAC	II/III	78%	73%	-	77%	*
		HGG			84%	100%	-	89%	*
		HGG			-	-	-	-	*
	1	HGG	BTV	0.55 cm^3^	98%	94%	0.955	98%	*
	1	HGG	BTV after 6 months	-					*
	1	LGG and HGG	TBR_mean_ + TBR_max_	-	66%	80%	0.78		NA
	1	HGG	TBR_mean_ + TBR_16mm_	-	75%	72%	-	74%	*
	1	HGG	TBR_max_ + TTP	2.85 and 35 min	36%	100%		70%	*
	3	LGG and HGG	TBR_mean_ + TTP	2.0 and/or 45 min	93%	100%		93%	*
		HGG		1.95 and 35 min	55%	100%		78%	*
		HGG		-	69%	78%	-	74%	*
	2	LGG and HGG	TBR_16mm_ + TTP	2.3 and/or 45 min	92%	73%		90%	*
		HGG		-	69%	83%		76%	*
	1	HGG	TBR_16mm/mean_ + TTP	1.95 and 32.5 min	89%	91%		90%	*
	1	HGG	TBR_max_+ TAC	2.3 and II/III	80%	91%		86%	*
	2	LGG and HGG	TBR_mean_ + TAC	2.0 and/or II/III	93%	73%		91%	*
		HGG		2.0 and II/III	60%	91%		76%	*
	1	LGG and HGG	TBR_16mm_ + TAC	2.3 and/or II/III	93%	73%		91%	*
	1	HGG	TBR_mean_ + Slope	-	50%	78%		65%	X
	2	LGG and HGG	TBR_16mm_ + Slope	1.95 and/or 0.69 SUV/h	96%	43%		86%	NA
		HGG		-	50%	89%		71%	*
	1	HGG	TBR_16mm/mean_ + Slope	1.95 and 0.32 SUV/h	78%	97%		93%	*
	1	HGG	TTP + Slope	-	56%	61%		59%	X
	1	HGG	TBR_16mm_ + TBR_mean_ + TTP	-	69%	89%		79%	*
	2	LGG and HGG	Radiomics features	-	73%	80%	0.85		NA
		HGG		-	100%	40%	0.74	70%	*
	1	LGG and HGG	TBR_max_ + TBR_mean_ + radiomics features	-	81%	70%	0.85		NA
Prognosis/Treatment response evaluation									
	1	LGG	Uptake	Positive vs. negative	-	-			X
	1	LGG and HGG	FET positivity	Yes vs. no	-	-			*
	1	HGG	New distant FET hotspot	Yes vs. no					*
	1	LGG and HGG	FET-PET response	Yes vs. no	-	-			*
	3	LGG and HGG	SUV_max_/BG	-	-	-			X
		LGG and HGG		-	-	-			X
		LGG and HGG		-	-	-			X
	1	LGG and HGG	Initial SUV_max_/BG	-	-	-			X
	2	LGG	SUV_mean_/BG	-	-	-			X
		HGG		-	-	-			X
	1	HGG	SUV_mean_/BG pre re-RT	2.2	-	-			X
	1	HGG	SUV_mean_/BG post re-RT	2.3	-	-			X
	1	LGG	SUV_total_/BG	-	-	-			X
	5	LGG and HGG	SUV_max_	2.3	-	-			X
		LGG and HGG			-	-			X
		LGG and HGG		2.6	-	-			*
		LGG and HGG		3.4	-	-			*
		HGG		Median	-	-			X
	1	HGG	SUV_mean_	Median	-	-			X
	1	HGG	SUV_peak_	Median	-	-			X
	12	LGG and HGG	TBR_max_	1.6	-	-			X
		LGG		2	-	-			X
		HGG		2.4	-	-			*
		LGG and HGG		2.5	-	-	0.696		*
		LGG and HGG		2.6	-	-			X
		HGG		3	-	-			X
		LGG and HGG		3.03	-	-			*
		HGG		Median (5.03)					*
		LGG		-	-	-			X
		LGG and HGG		-	-	-			X
		LGG and HGG		-	-	-			X
		HGG		-	-	-			X
	1	HGG	TBR_max-preRCx_	2.9 (OS)	68%	73%			*
	2	LGG and HGG	TBR_max_ baseline	2.0 (PFS)/1.9 (OS)	-	-			* (PFS)
		HGG		-	-	-			NA
	1	LGG and HGG	TBR_max_ after 6 months	-	-	-			X
	1	HGG	Early TBR_max_ response	∆-10%	-	-			*
	1	LGG and HGG	TBR_max_ response	∆ ± 10%	-	-			*
	3	LGG and HGG	TBR_max_ change	0%	-	-			*
		HGG			-	-			*
		HGG		∆-10% (PFS)/∆-20% (OS)	83% (OS)	67% (OS)	0.75 (OS)		*
	1	HGG	TBR_max_ pre re-RT	3.3	-	-			X
	1	HGG	TBR_max_ post re-RT	2.6	-	-			X
	1	HGG	TBR_16mm_ relative reduction	27%	92%	63%	0.78		NA
	1	HGG	∆TBR_max_ scan 2	-	-	-			NA
	1	HGG	∆TBR_max_ scan 3	-	-	-			NA
	2	HGG	TBR_mean_	2	-	-			*
		HGG		2.05	60%	70%	0.69		X
	1	HGG	TBR_mean_ relative reduction	16%	92%	63%	0.81		NA
	1	LGG and HGG	TBR_16mm_ baseline	1.9 (PFS)/1.8 (OS)	-	-			X
	3	LGG and HGG	TBR_16mm_ change	0%	-	-			X
		HGG			-	-			X
		HGG		∆-5%	67%	75%	0.72		*
	1	HGG	TBR in non-contrast enhancing tumor portions at follow-up	High vs. low	-	-			*
	1	LGG	TBR_3SD_	1.1	-	-			*
	1	LGG and HGG	TBR_10mm_	2.3	-	-	0.696		*
	1	HGG	TBR_16mm_	2.55	70%	57%	0.63		X
	5	HGG	TBR	1.56 (PFS)/1.57 (OS)	-	-			X
		HGG		2	-	-			X
		HGG		2.4	-	-			*
		HGG		2.5	-	-			X
		HGG		Median	-	-			X
	1	HGG	Early TBR response	∆-10%					*
	1	HGG	TLG	Median					X
	1	HGG	TLU	35.0 (PFS)/17.1 (OS)	-	-			* (OS)
	1	LGG and HGG	TTP	20 min	-	-	0.848		*
	1	HGG		25 min	90%	87%	0.90		*
	1	LGG and HGG		-	-	-			*
	1	HGG	TTP change	0%	-	-			X
	1	HGG	TTP_min_	12.5 min	-	-			*
	1	LGG and HGG		>25 min vs. 12.5 < t ≤ 25 min vs. ≤12.5 min	-	-			*
	1	LGG and HGG		17.5 min	-	-			*
	1	HGG	Slope	−0.103 SUV/h	70%	90%	0.77		X
	1	LGG and HGG		-	-	-			*
	1	LGG and HGG	Slope-to-peak	7 × 10^−5^/s	-	-	0.711		*
	5	LGG	TAC	Increasing vs. decreasing	-	-			*
		LGG and HGG		Homogeneous increasing vs. mixed vs. homogeneous decreasing	-	-			*
		LGG and HGG		Homogeneous decreasing vs. focal decreasing vs. homogeneous increasing	-	-			*
		HGG		Increasing	-	-			*
		LGG and HGG		-	-	-			*
	1	HGG	TAC pre re-RT	G_1–2_ vs. G_3_ vs. G_4–5_					*
	1	HGG	TAC post re-RT	G_1–2_ vs. G_3_ vs. G_4–5_					X
	1	LGG and HGG	Initial TAC	Increasing vs. decreasing	-	-			X
	1	HGG		Increasing vs. decreasing (OS)	-	-			*
	1	LGG and HGG	TAC after 6 months	Increasing vs. decreasing	-	-			X
	1	LGG and HGG	TAC response	Stable increasing vs. decreasing to increasing vs. Increasing to decreasing vs. Stable decreasing	-	-			X
	1	LGG and HGG	Peak TBR	2.2	-	-	0.704		*
	8	HGG	BTV	4.3 cm^3^	-	-			*
		LGG and HGG		10 cm^3^					*
		HGG		11.15 mL	72%	54%	0.56		X
		HGG		19.4 (PFS)/18.9 (OS)	-	-			* (OS)
		HGG		20 mL	-	-			*
		HGG		Median					X
		LGG		-	-	-			X
		HGG		-	-	-			X
	1	HGG	BTV_preRCx_	9.5 cm^3^	64%	70%			*
	1	LGG and HGG	Initial BTV	-	-	-			X
	1	LGG and HGG	BTV baseline	28.2 mL (PFS)/13.8 mL (OS)	-	-			*
	1	HGG		-	-	-			*
	1	LGG and HGG	BTV after 6 months	-	-	-			*
	1	HGG	Absolute BTV at follow-up	5 mL	85%	88%	0.92		*
	1	LGG and HGG	BTV response	∆ ± 25%	-	-			*
	3	LGG and HGG	BTV change	0%	-	-			*
		HGG		0%	-	-			*
		HGG		0% (PFS)	-	-	-		*
	1	HGG	BTV relative reduction	27%	77%	63%	0.82		NA
	1	HGG	∆BTV scan 2	-	-	-			X
	1	HGG	∆BTV scan 3	-	-	-			X
	1	LGG and HGG	BTV_SUVmax≥2.2_	-	-	-			X
	1	LGG and HGG	BTV_≥40%SUVmax_	-	-	-			X
	1	HGG	BTV_TBR≥ 1.6_	25 mL	-	-			*
	1	HGG	BTV_TBR≥ 2.0_	10 mL	-	-			*
	1	HGG	BTV pre re-RT	13.7 cc	-	-			X
	1	HGG	BTV post re-RT	7.3 cc	-	-			X
			Radiomic features:						*
	1	HGG	SUV_min_	-	-	-			*, &
	1	HGG	SUV_mean_	-	-	-			*, &
	1	HGG	GLV	-	-	-			*, &
	1	HGG	GLV2	-	-	-			*, &
	1	HGG	WF_GLV	-	-	-			*, &
	1	HGG	Qacor	-	-	-			*, &
	1	HGG	QHGZE	-	-	-			*, &
	1	HGG	QSZHGE	-	-	-			*, &
	1	HGG	QGLN2	-	-	-			*, &
	1	HGG	QHGRE	-	-	-			*, &
	1	HGG	QSRHGE	-	-	-			*, &
	1	HGG	QLRHGE	-	-	-			*, &
	1	HGG	SZLGE	-	-	-			*
	1	HGG	Busyness	1.366 (PFS)/0.984 (OS)	-	-			*
	1	HGG		-	-	-			*, &
	1	HGG	WF_TS	-	-	-			*, &
	1	HGG	QvarianceCM	-	-	-			*, &
	1	HGG	Eccentricity	-	-	-			*, &
	1	HGG	Coarseness	5.96 × 10^−3^ (PFS)/6.88 × 10^−3^ (OS)	-	-			*
	1	HGG	Contrast	0.427	-	-			*
	1	HGG	Complexity	0.085 (PFS)/0.094 (OS)	-	-			*
	1	HGG	SUV_mean_ + WF_GLV + QLRHGE + SUV_min_	-	-	-			*
	1	HGG	SZLGE + Busyness + QVarianceCM + Eccentricity	-	-	-			*

## Data Availability

No new data were created or analyzed in this study. Data sharing is not applicable to this article.

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
