# Peer review of "Contribution of [18F]FET PET in the Management of Gliomas, from Diagnosis to Follow-Up: A Review"

_pharmaceuticals, 2024, doi:10.3390/ph17091228_

Round 1

Reviewer 1 Report

Comments and Suggestions for Authors

This is an excellent overview on the application of [18F]FET for positron emission tomography (PET) in the detection, diagnosis and follow-up of gliomas. It is recommended to be accepted for publication after some revision on the basis of comments below.

COMMENTS

1.

Although the chemical structures and metabolic pathways of the labelled amino acids used as PET imaging agents are displayed in Figure 1, a detailed description of the mechanism and advantages/disadvantages of these compounds in comparison to each other as PET contrasting agents in relation to the detection and diagnosis of gliomas should be described in details in this review.

2.

The captions for Tables 1 and 2 are suggested to be written at the top of these Tables.

3.

Tables are recommended to be rearranged in full page width for better clarity and for better information quality.

4.

Scheme 1 (as Supplementary Table 1) is not necessary to be included separately in the Supplementary. This should be included at either before Tables 1 and 2, and described, or at the end of the manuscript and referred to and described it in the text accordingly.

Author Response

Comment 1 : Although the chemical structures and metabolic pathways of the labelled amino acids used as PET imaging agents are displayed in Figure 1, a detailed description of the mechanism and advantages/disadvantages of these compounds in comparison to each other as PET contrasting agents in relation to the detection and diagnosis of gliomas should be described in details in this review.

Response 1 : That is right. We added a description of the different radiolabeled amino acids before Figure 1.

Comment 2 : The captions for Tables 1 and 2 are suggested to be written at the top of these Tables.

Response 2 : Thanks, we changed it.

Comment 3 : Tables are recommended to be rearranged in full page width for better clarity and for better information quality.

Response 3 : Thanks for pointing this out, we fixed the tables presentation and repeated the first line on other pages to ensure better reading.

We initially submitted it in a landscape orientation to ease these issues and have to point that the column "accuracy" had been deleted somewhere in the process, we corrected it.

Comment 4 : Scheme 1 (as Supplementary Table 1) is not necessary to be included separately in the Supplementary. This should be included at either before Tables 1 and 2, and described, or at the end of the manuscript and referred to and described it in the text accordingly.

Response 4 : Agree. We included it as Table 3 before the PET-only table since it reports the homogenization work we did between Tables 1 and 2 (now Table 2 and 4). We also reworded differently the introduction of tables 3 and 4 in the results.

Additional modification :

We found an error in the 3.10. section : there were 1141 tumors included in this indication and 992 of them were HGG. We initially counted 1045 and 901, respectively. We corrected it.

Reviewer 2 Report

Comments and Suggestions for Authors

Diagnosis and treatment of gliomas, the most prevalent kind of primary malignant brain tumors in adults, are significantly complicated by their variety and potential aggressiveness. The purpose of this review is to assess the effectiveness of O-(2-[18F]fluoroethyl)-L-tyrosine ([18F]FET) positron emission tomography (PET) as an imaging technique for improving the clinical treatment of gliomas. The results validate that [18F]FET is effective in precisely defining tumor tissue, enhancing diagnostic precision, and assisting in therapeutic decision-making by offering vital information on tumor metabolism. The results are interesting. I found the paper well written, clear and of direct relevance for technicians working for local authorities which have to take such important decisions. The subject addressed is within the scope of the journal. The paper will be more interesting if the authors address the following comments.

  1. Please revise the abstract to present it in a single paragraph.
  2. Consider including a flowchart of the study to clarify the research process.
  3. The materials and methodology section should be elaborated to provide more detailed information.
  4. Please include the mathematical equations for accuracy, sensitivity, and specificity to support the findings.
  5. Clearly identify the specific contributions of the study. Additionally, discuss the potential application of the proposed methods to other problems, and include statements regarding future research directions.
Comments on the Quality of English Language

Minor editing of English language required.

Author Response

Comment 1 : Please revise the abstract to present it in a single paragraph.

Response 1 : Thanks, we corrected it.

Comment 2 : Consider including a flowchart of the study to clarify the research process.

Response 2 : Thanks for your comment, a flowchart is presented in Figure 2.

Comment 3 : The materials and methodology section should be elaborated to provide more detailed information.

Response 3 : Thanks, we added more information.

Comment 4 : Please include the mathematical equations for accuracy, sensitivity, and specificity to support the findings.

Response 4 : We're not sure we quite understand the requirement. Do you want us to remind the formulas to obtain these parameters ? We used the data provided by the articles and didn't calculate any of these parameters.

Comment 5 : Clearly identify the specific contributions of the study. Additionally, discuss the potential application of the proposed methods to other problems, and include statements regarding future research directions.

Response 5 : We specifically noticed contributions fo the study at the begining of the discussion section as following :  « This review propose an up-to-date summary of PET performance in glioma management using O-(2-[18F]fluoroethyl)-L-tyrosine. The homogenization of PET tumor-to-brain ratios according to the determination of the different region-of-interest allowed to truly compare their sensibility, specificity, AUC and accuracy. » Regarding future research, we think that the following sentence at the end of the discussion section is valuable : « There is a need to pursue research with prospective, multicentric studies to be able to standardize imaging analysis and define the use of technological advancements such as hybrid PET/MRI imaging and radiomics, and to compare [18F]FET with existing radiopharmaceuticals such as [18F]F-DOPA head-to-head comparisons.”